
# An improved method for atmospheric [14]CO measurements

Vasilii V. Petrenko[1], Andrew M. Smith[2], Edward M. Crosier[1], Roxana Kazemi[1], Philip Place[1], Aidan Colton[3], Bin Yang[2], Quan Hua[2] and Lee T. Murray[1]

[1]Department of Earth and Environmental Sciences, University of Rochester, Rochester, NY 14627, USA
[2]Australian Nuclear Science and Technology Organisation (ANSTO), Locked Bag 2001, Kirrawee DC, NSW 2232, Australia.
[3]NOAA Earth System Research Laboratory, Global Monitoring Division, Boulder, Colorado, USA.

*Correspondence to*: Vasilii V. Petrenko (vasilii.petrenko@rochester.edu)

## Abstract

Important uncertainties remain in our understanding of the spatial and temporal variability of atmospheric hydroxyl radical concentration ([OH]). Carbon-14-containing carbon monoxide ([14]CO) is a useful tracer that can help in the characterization of [OH] variability. Prior measurements of atmospheric [14]CO concentration ([[14]CO] are limited in both their spatial and temporal extent, partly due to the very large air sample volumes that have been required for measurements (500 – 1000 liters at standard temperature and pressure, L STP) and the difficulty and expense associated with the collection, shipment and processing of such samples. Here we present a new method that reduces the air sample volume requirement to ≈90 L STP while allowing for [[14]CO] measurement uncertainties that are on par with or better than prior work (≈3 % or better, 1 σ). The method also for the first time includes accurate characterization of the overall procedural [[14]CO] blank associated with individual samples, a key improvement over prior atmospheric [14]CO work. The method was used to make measurements of [[14]CO] at the NOAA Mauna Loa Observatory, Hawaii, USA, between November 2017 and November 2018. The measurements show the expected [[14]CO] seasonal cycle (lowest in summer) and are in good agreement with prior [[14]CO] results from another low-latitude site in the Northern Hemisphere. The lowest overall [[14]CO] uncertainties (2.1 %, 1 σ) are achieved for samples that are directly accompanied by procedural blanks and whose mass is





increased to ≈ 50 micrograms of carbon (μgC) prior to the $^{14}$C measurement via dilution with a high-CO, $^{14}$C-depleted gas.


# 1 Introduction

## 1.1 The importance of improving the understanding of OH variability

Atmospheric hydroxyl radical concentration ([OH]) is arguably the single most important parameter in characterizing the overall chemical state of the atmosphere because OH serves as the main atmospheric oxidant. Reaction with OH removes a large number of atmospheric trace species, including reactive greenhouse gases like methane as well as most anthropogenic pollutants (e.g., Brasseur et al., 1999). Changes in [OH] in space and

time impact both global air quality and the rate of climate change. While our understanding of and ability to predict global OH abundance and variability continues to improve, large uncertainties remain. This was highlighted, for example, by the Atmospheric Chemistry and Climate Modeling Intercomparison Project (ACCMIP), where individual models disagreed by ± 50 % in their calculations of global mean [OH]

(Naik et al., 2013; Voulgarakis et al., 2013).

OH is very short-lived (lifetimes of 1 s or less are typical) and heterogeneously distributed (e.g., Spivakovsky et al., 2000), making measurements inherently challenging. Therefore, characterizing global mean [OH] via direct measurements is not feasible.

Instead, a number of tracers have been used for this purpose, including $^{14}$CO (e.g., Brenninkmeijer et al., 1992), methane (CH$_4$; Montzka et al., 2011), methyl chloroform (MCF; CH$_3$CCl$_3$; e.g., Montzka, et al., 2011; Prinn et al., 2001), as well as a combination of hydrofluorocarbons (HFCs) and hydrochlorofluorocarbons (HCFCs) (Liang et al., 2017). The approach involves selecting a trace gas with a well-characterized source and

with OH as the dominant sink.

Over the last ≈2 decades, the most reliable characterization of global mean [OH] has been derived from MCF (e.g., Montzka, et al., 2011; Prinn, et al., 2001). However, MCF



atmospheric mixing ratios have been declining rapidly as a result of phase-out of its

production. This makes the continued use of MCF for studies of [OH] challenging, as MCF mixing ratios approach analytical detection limits and as estimates of [OH] become increasingly sensitive to poorly-characterized residual MCF emissions (e.g., Rigby et al., 2017). Furthermore, while the moderately long lifetime of MCF (≈5 years; Rigby et al., 2013) has allowed for constraints on global and hemispheric mean [OH], less is known

about [OH] temporal and spatial variability, which is critical for understanding the evolution, transport and fate of air pollutants.

**1.2 $^{14}$CO as a tracer for atmospheric OH**

Evidence from measurements of carbon-14 of atmospheric carbon monoxide ($^{14}$CO) provided the first indication that carbon monoxide had a relatively short atmospheric lifetime, leading to the suggestion that tropospheric OH may be important in the removal of CO (Weinstock, 1969). Since then, measurements of $^{14}$CO concentration ([$^{14}$CO]) have been used by several research groups to improve understanding of tropospheric [OH]

(e.g., Brenninkmeijer, et al., 1992; Jöckel and Brenninkmeijer, 2002; Manning et al., 2005; Quay et al., 2000; Volz et al., 1981).

$^{14}$CO has a strong, reliable and well-characterized primary source. This is an advantage over CO, CH$_4$, or halocarbon tracers for OH, which typically have variable emissions that

are associated with relatively large uncertainties. $^{14}$C is produced from $^{14}$N via interactions with neutrons ($^{14}$N(n,p)$^{14}$C) resulting from bombardment of the atmosphere by galactic cosmic rays. Production rates are highest in the upper troposphere and lower stratosphere (UT/LS), with about half of $^{14}$C produced in each region. The geomagnetic field provides the strongest cosmic ray shielding in the low latitudes, resulting in higher

$^{14}$C production rates in the mid- and high latitudes (e.g., Masarik and Beer, 1999). Variations in the $^{14}$C production rate are well-characterized from neutron monitor observations (e.g., Kovaltsov et al., 2012). Once produced, $^{14}$C quickly reacts to form $^{14}$CO, with ≈93% yield (Mak et al., 1994).



The dominant $^{14}CO$ removal mechanism is via reaction with OH; $^{14}CO$ can therefore in principle serve as a tracer for OH abundance and variability. There are several aspects of atmospheric cycling of $^{14}CO$ that offer either challenges or advantages in its use as a tracer for [OH], depending on the question being posed. First, $^{14}CO$ (and CO) has a relatively short average tropospheric lifetime of $\approx 2$ months, which varies by latitude

(shortest in the tropics) and by season (shortest in season of maximum insolation), following variations in [OH] (e.g., Spivakovsky, et al., 2000). This is much shorter than the interhemispheric mixing time of $\approx 1$ year, and means that [$^{14}CO$] measurements at a given station are sensitive to regional rather than global [OH] (Krol et al., 2008). The fact that [$^{14}CO$] measurements at a given station are mainly sensitive to [OH] in a spatially

limited region presents a challenge for using [$^{14}CO$] to constrain global mean [OH] abundance and variability. To ensure robust characterization of global mean [OH] from [$^{14}CO$] alone, records for multiple sampling stations are necessary.

         The limited spatial footprint of [$^{14}CO$] sensitivity to [OH] can instead be an advantage if

the question is one of OH spatial and seasonal variability. Driven by strong seasonality and meridional gradients in [OH], cosmogenic production rates, and stratosphere-to-troposphere (STT) transport, as well as a relatively short chemical lifetime, [$^{14}CO$] near the surface shows strong seasonal and meridional variability (e.g., Jöckel and Brenninkmeijer, 2002).


**1.3 Atmospheric [$^{14}CO$] measurement techniques and associated challenges**

$^{14}CO$ is an ultra-trace constituent of the atmosphere, with surface concentrations ranging between $\approx 4 - 25$ molecules / cm$^3$ STP. This has necessitated very large sample volumes

of $500 - 1000$ L STP for the analyses (e.g., Brenninkmeijer, 1993; Mak, et al., 1994). Air samples are typically collected into high-pressure aluminum cylinders with the use of modified 3-stage oil-free compressors (e.g., Mak and Brenninkmeijer, 1994). The collected air is processed by first removing condensable gases using high-efficiency cryogenic traps (Brenninkmeijer, 1991), followed by oxidation of CO to $CO_2$ using the

Schutze reagent and subsequent cryogenic trapping of the CO-derived $CO_2$ using liquid

nitrogen (Brenninkmeijer, 1993). The produced $CO_2$ is then graphitized and analyzed for $^{14}C$ using accelerator mass spectrometry (AMS) (Brenninkmeijer, 1993).

There are two main challenges associated with atmospheric $^{14}CO$ measurements. First, the very large air sample volumes and the need for high-pressure gas cylinders result in

relatively complex and expensive logistics and sample processing. These challenges have limited the extent of $^{14}CO$ atmospheric measurements collected to date. Second, $^{14}CO$ production by cosmic rays via the $^{14}N(n,p)^{14}C$ mechanism continues in air sample containers after the samples have been collected (the "*in situ* component"; e.g., Lowe et al., 2002; Mak et al., 1999). This effect is particularly large for samples stored at high

altitudes / latitudes, as well as for samples transported by air, and has contributed significantly to uncertainties in interpretation of $[^{14}CO]$ measurements (e.g., Jöckel and Brenninkmeijer, 2002).

In this paper, we describe a new method for atmospheric $[^{14}CO]$ measurements that

addresses both of the above challenges, demonstrate the use of this method, and discuss how measurement uncertainties can be minimized in this approach.

## 2 New method for smaller-sample atmospheric $^{14}CO$ measurements


### 2.1 Atmospheric sample collection system and procedure

The new atmospheric sampling system (Figure 1) was developed and installed at the NOAA Mauna Loa observatory (MLO; 19.5˚N, 155.6˚W, 3397 m above sea level) in

November 2017. A 3/8" OD inlet line (Synflex 1300) was mounted near the top of a ≈36 m tower. A small diaphragm pump (Air Cadet EW-07532-40) continuously flushes the inlet line at a flow rate of ≈5 LPM when not sampling. The main part of the sampling system consists of a drying trap (45 g of anhydrous $Mg(ClO_4)_2$ in a 1" OD steel tube), a CO removal trap (25 g of  Sofnocat 423 from Molecular Products in a ½" OD steel tube),

a diaphragm compressor (KNF N145 with neoprene diaphragms) and a pre-evacuated lightweight electropolished stainless steel canister (Essex Cryogenics, 35 L internal volume).



Prior to collecting an air sample, the diaphragm compressor is leak-checked using the pressure gauge. The air flow is then started into the main part of the system and bypasses the Sofnocat CO scrubber; the flow is adjusted to $\approx$ 5 LPM using the metering valve. The system is flushed for 4 min; then the connection to the sample canister is pressure-flushed 3 times. The sample canister is initially opened slowly, keeping the pressure upstream of the canister slightly above ambient (to minimize the impact of any leaks and help maintain a relatively constant flow rate); then opened fully once pressure in the canister reaches ambient.

In an attempt to provide some temporal averaging for $^{14}CO$ samples at MLO, most sample canisters were filled in 2 separate sessions $\approx$1 week apart, with half the air volume collected each time. A few of the canisters (Table S1) were filled in a single session, when atmospheric conditions at MLO did not allow for sampling during one of the targeted weeks (e.g., during volcanic plumes). The final air volumes in the canisters were $\approx$ 90 L STP, allowing for non-hazardous shipping. The system also allows for air collection in blank mode, where the flow is directed through the Sofnocat CO scrubber. This removes all $^{14}CO$ (and CO), allowing to assess the cumulative procedural addition of extraneous $^{14}CO$ to the samples, including in situ $^{14}CO$ production by cosmic rays inside the canisters during transport and storage. Samples were collected between November 2017 and November 2018. Every 2 weeks, 2 canisters were filled: either 2 samples, or a sample and a blank (Tables S1 and S2). Once complete, sample and blank canisters were moved down to sea level on the same day to minimize in situ $^{14}CO$ production (which increases approximately exponentially with altitude in the troposphere) and shipped via air to the University of Rochester within $1 - 2$ days.

## 2.2 Sample air processing and measurements

Sample air processing and measurement approaches at U Rochester are based on methods developed earlier for $^{14}CO$ analyses in samples of air extracted from glacial firn and ice (Dyonisius et al., 2020; Hmiel et al., 2020; Petrenko et al., 2016; Petrenko et al., 2017). Here we provide a brief description, including changes and details specific to the MLO $^{14}CO$ samples. The air samples are first measured for CO mole fraction ([CO]) against





NOAA-calibrated standards using a Picarro G2401 cavity ring-down spectroscopic analyzer. A high-[CO] gas ($10.02 \pm 0.06$ µmol mol[-1]) containing $^{14}$C-depleted CO is then added to the sample canisters; this step will henceforth be referred to as the "dilution". The dilution simultaneously serves to increase the carbon mass in the sample to a level

that is necessary for robust measurement by AMS and reduce the $^{14}$C activity of the samples to values that are within the range of common $^{14}$C measurement standards.

The relative proportions of sample air and the high-[CO] dilutant gas are determined using a Paroscientific 745-100A pressure transducer (0.01% absolute accuracy) while monitoring the canister temperatures. For the first $\approx 2/3$ of the samples, the dilutions were

designed to produce a final sample size of $\approx 22$ micrograms of carbon (µgC). For the final $\approx 1/3$ of the samples, the amount of the dilutant gas was increased to produce final sample sizes of $\approx 50$ µgC, to investigate whether the somewhat larger sample sizes would yield smaller overall uncertainties.

The diluted air samples were processed using a system previously developed at U Rochester (Dyonisius, et al., 2020; Hmiel, et al., 2020). Briefly, the sample air stream (at 1 LPM STP) first passes through a coaxial Pyrex trap held at -75˚C, followed by four Pyrex traps containing nested fiberglass thimbles ("Russian Doll" traps; Brenninkmeijer, 1991) held at -196˚C with liquid nitrogen. These traps serve to remove $H_2O$, $CO_2$ and

other condensable gases. The Russian Doll traps are also very effective at removing hydrocarbons, including C2 hydrocarbons (Brenninkmeijer, 1991; Petrenko et al., 2008; Pupek et al., 2005). Following cryogenic purification, the air stream passes through a furnace containing 2 g of platinized quartz wool held at 175˚C; this oxidizes CO to $CO_2$ while allowing $CH_4$ to pass through unaffected. The CO-derived $CO_2$ is then

cryogenically trapped and further purified to remove trace amounts of $H_2O$ and air. The amount of collected $CO_2$ is then quantified in a calibrated volume, and the $CO_2$ is flame-sealed into 6 mm OD Pyrex tubes for storage and shipment to the AMS facility. This $CO_2$ is converted to graphite (Yang and Smith, 2017) and subsequently measured for $^{14}$C using the 10 MV ANTARES accelerator facility at ANSTO (Smith et al., 2010). The MLO

samples and blanks were processed at ANSTO in four separate sets, and each of these sets was accompanied by commensurately-sized $^{14}$C standards and blanks prepared at


ANSTO, including the international [14]C standards HOxII, IAEA-C7, IAEA-C8, and aliquots from a previously well-characterized cylinder of [14]C-depleted $CO_2$.

230 $\delta^{13}C$ of CO in the high-[CO] [14]C-depleted dilution gas (needed for [14]C normalization; e.g., Stuiver and Polach, 1977) was measured as described in Dyonisius, et al. (2020). $\delta^{13}C$ of CO in the air samples was measured using a new system at the University of Rochester, following the design and procedure described in Vimont et al. (2017).

235 **2.3 Data processing and corrections**

The data processing and corrections approach largely follows prior work (e.g., Dyonisius, et al., 2020; Petrenko, et al., 2016). Here we provide a brief summary as well as highlight differences from prior work. First, in a departure from prior work, measured [14]C values

240 (in pMC units; Stuiver and Polach, 1977) are empirically corrected for any effects of processing at ANSTO (handling of sample-derived $CO_2$, conversion to graphite and the AMS measurement). This is accomplished by plotting the measured [14]C values of commensurately-sized standards against the accepted [14]C values for these standards, and using the Igor Pro software to determine linear fit coefficients and associated

245 uncertainties (Fig. 2). This correction was determined separately for each measured set of MLO samples and blanks, and is small (<2% in all cases).

[CO] in the diluted samples and blanks was calculated based on [CO] in the samples and in the high-[CO] dilution gas and the pre- and post-dilution pressures, corrected for any

250 temperature change in the canisters in between the two pressure measurements. $\delta^{13}C$ of CO in the diluted samples was calculated using an equivalent approach. [14]CO content in the diluted samples and blanks is then calculated using:

$$^{14}C = \frac{pMC}{100} \times e^{-\lambda(y-1950)} \times \frac{\left(1+\frac{\delta^{13}C}{1000}\right)^2}{0.975^2} \times 1.1694 \times 10^{-12} \times [CO] \times \frac{1}{22400} \times N_A \qquad (1)$$

where [14]C is the number of [14]CO molecules per $cm^3$ STP, pMC is the measured sample or

255 blank [14]C activity in pMC units after the empirical correction for ANSTO processing, $\lambda$ is the [14]C decay constant (1.210 x $10^{-4}$ $yr^{-1}$), y is the year of measurement, $\delta^{13}C$ is the calculated $\delta^{13}C$ of CO in the diluted sample or blank, 0.975 is a factor arising from [14]C

activity normalization to $\delta^{13}C$ of -25 ‰ associated with pMC units, $1.1694 \times 10^{-12}$ is the $^{14}C / (^{13}C + ^{12}C)$ ratio corresponding to the absolute international $^{14}C$ standard activity

(Hippe and Lifton, 2014), 22400 is the number of $cm^3$ STP of gas per mole, and $N_A$ is the Avogadro constant.

Next, the $^{14}CO$ content in the diluted samples and blanks that is attributable to the high-[CO] $^{14}C$-depleted dilution gas is calculated, again using Equation 1. Triplicate aliquots

of dilution gas (all $\approx 50$ μgC) were processed and measured for $^{14}C$ near the start and again at the end of the 1-year sampling campaign. The $^{14}C$ activity of CO in the dilution gas is expected to increase slowly with time due to in situ production in the gas cylinder. For the analysis of the first MLO sample set, the mean value obtained from the initial set of $^{14}C$ measurements of the dilution gas was used ($0.19 \pm 0.04$ pMC, $1\sigma$, after corrections

for ANSTO processing). For the analysis of the final MLO sample set, the mean value obtained from the second set of $^{14}C$ measurements of the dilution gas was used ($0.46 \pm 0.10$ pMC). For the analysis of the second and third MLO sample sets, the average of the two sets of $^{14}C$ measurements on the dilution gas was used. For the $^{14}CO$ content calculation in this case, [CO] is the CO mole fraction in the diluted samples and blanks

that is attributable to the dilution gas only.

The $^{14}CO$ content that is attributable to the high-[CO] $^{14}C$-depleted dilution gas is then subtracted from the total $^{14}CO$ content. The $^{14}CO$ content is then further corrected for the volumetric effect of the dilution, which reduces the number of $^{14}CO$ molecules per $cm^3$

STP of gas. This yields the $^{14}CO$ content in undiluted samples and blanks. The final step of the data processing involves the procedural blank correction. For samples that were directly accompanied by a blank, the $^{14}CO$ content of that blank is subtracted. This accounts for all extraneous $^{14}CO$ affecting that particular sample. For samples that were not directly accompanied by a blank, the average $^{14}CO$ content determined from all

blanks collected in a similar mode (tanks filled on 2 separate days $\approx 1$ week apart versus tanks filled in a single session) was subtracted.



All uncertainties were propagated through the data reduction / correction calculations using standard error propagation techniques. For one of the sample sets, the errors were also propagated using a Monte Carlo approach to confirm that this yields equivalent uncertainties.

## 3 Results and Discussion

The MLO sample and blank [$^{14}$CO] results are shown in Figure 3 and listed in Tables S1 and S2. [$^{14}$CO] at MLO during the year of sampling ranged from 5 – 13 molecules per cm$^3$ STP. There is a clear seasonal cycle, with lowest values during the summer and highest values during the winter, as observed in prior work (e.g., Manning, et al., 2005). The relatively high temporal variability in [$^{14}$CO], which is particularly prominent in the winter season, is likely driven by the competing influences of low-latitude versus mid-latitude air masses at MLO ([$^{14}$CO] shows a very strong meridional gradient, particularly in the winter season, with much higher values at higher latitudes; e.g. Jöckel and Brenninkmeijer, 2002). For a first-order comparison with prior [$^{14}$CO] measurements we consider Ragged Point, Barbados (13.2˚N), which is the station with available finalized and previously published [$^{14}$CO] measurements that is closest in latitude to MLO (19.5˚N). The prior Barbados [$^{14}$CO] measurements (July 1996 - July 1997; Mak and Southon, 1998) showed seasonal [$^{14}$CO] variability in a similar range (5 – 12 molecules per cm$^3$ STP) as our new MLO data, although the Barbados measurements were not corrected for in situ $^{14}$CO production in the sample tanks and atmospheric $^{14}$C production may have been somewhat different during 1996 -1997 as compared to 2017 - 2018.

The average 1 σ overall uncertainty of the measured MLO [$^{14}$CO] values after corrections (obtained via uncertainty propagation) is 0.27 molecules per cm$^3$ STP, or 3.3% of the average [$^{14}$CO] value. Pooled standard deviation computed from 12 replicate sample pairs provides an estimate of repeatability and is 0.18 molecules per cm$^3$ STP, corresponding to 2.2% of the average $^{14}$CO value for all the replicate samples. MLO is a low-latitude site, with lower [$^{14}$CO] as compared to most previously-monitored sites; this means that the same absolute [$^{14}$CO] uncertainty would translate into a larger relative uncertainty for

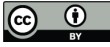

MLO than for most other sites. Despite this, our results compare well with overall 1 σ uncertainties reported in prior work that used much larger samples at sites with higher [$^{14}$CO] (4% for Quay, et al., 2000 and 4 – 5% for Manning, et al., 2005). Brenninkmeijer (1993) and Röckmann et al. (2002) report [$^{14}$CO] uncertainties of ≈2%, but those estimates did not take into account the uncertainty associated with the correction for in

situ $^{14}$CO production in sample tanks during storage and transport.

The overall procedural blank for the MLO $^{14}$CO samples (Fig. 3; Table S2) is due almost entirely to $^{14}$CO production in sample canisters during storage and transport. This blank is relatively large (average blank [$^{14}$CO] amounts to 16% of the average corrected sample

[$^{14}$CO]) and variable (relative standard deviation of 21%), highlighting the need for accurate blank characterization. In situ $^{14}$CO production in the sample canisters during storage at the high altitude MLO site in between the two days on which the canisters are filled and during aircraft transport from Hawaii to Rochester both appear to be important. Two of the blank canisters were filled in a single day, rather than half-filled on two

separate days a week apart (Table S2). For these two blanks, average [$^{14}$CO] is 0.95 molecules per cm$^3$ STP, as compared to average [$^{14}$CO] of 1.42 molecules per cm$^3$ STP for the ten blanks half-filled on two separate days.

One of the main objectives with the MLO sample set was method optimization to reduce

uncertainties. We used a two-sample t-test to investigate the effects of sample carbon mass and whether or not a sample was directly accompanied by a procedural blank on the overall sample [$^{14}$CO] uncertainties after corrections (Table 1). A procedural blank that directly accompanies a sample should in principle be affected by the same amount of in situ $^{14}$CO production, allowing for the blank $^{14}$CO content to be directly subtracted from

the $^{14}$CO content of the accompanying sample. For samples that are not directly accompanied by a blank, the variability in the blanks must be considered, adding to uncertainty. As expected, the overall uncertainties are significantly lower for samples that are accompanied by blanks (Table 1). This finding is true if all samples are considered, as well as for the ≈22 μgC and ≈50 μgC sample subsets.


Sample carbon mass (mass of graphite actually measured for [14]C by AMS) may matter for two reasons. First, a larger carbon mass in principle makes the sample less susceptible to problems during graphitization and AMS measurement. Second, an analysis of the relative contributions of individual uncertainties to the final overall uncertainty revealed

that the uncertainty arising from the dilution with the high-[CO] [14]C-depleted gas was a key contributor. For the smaller ≈22 µgC final sample masses, a relatively small amount of the high-[CO] gas (≈4 L STP) was being added to a large amount of sample air (≈90 L STP). This resulted in a relative error of ≈2% for the fraction of the diluted sample carbon that originated from the high-[CO] gas. Increasing the final sample carbon mass to

≈50 µgC via increasing the amount of the high-[CO] gas added during dilution reduces this relative error to < 1%. Surprisingly, we did not observe a significant reduction in the relative [[14]CO] uncertainty when all ≈22 µgC samples are compared to all ≈50 µgC samples (Table 1). However, there was a significant uncertainty reduction associated with larger sample mass if only the subset of samples directly accompanied by blanks was

considered.

**Conclusions**

The described new atmospheric [[14]CO] measurement method uses much smaller sample air volumes than prior work, simplifying sample collection, processing and field logistics and reducing costs; the new method appears to perform well. The MLO [[14]CO] measurements made with this method show good first-order agreement with prior measurements at a different Northern Hemisphere low latitude site. The method allows

for accurate characterization of the extraneous [14]CO component from in situ cosmogenic production in sample canisters, showing that this component can be relatively large and variable. In terms of sample measurement uncertainties, the new method compares favorably with prior work that utilized 5 – 10 times larger air sample volumes. A significant improvement in overall measurement uncertainties is achieved for samples

that are directly accompanied by procedural blanks, highlighting the usefulness of this mode of sample collection. The lowest overall [[14]CO] uncertainties (2.1 %, 1 σ) were achieved for samples that were directly accompanied by procedural blanks and were



diluted with a relatively larger amount of high-[CO] $^{14}$C-depleted gas to increase the final sample sizes for AMS analysis to ≈ 50 μgC.


### Data availability

All the new [$^{14}$CO] data discussed in this manuscript are available in the Supplement (Tables S1 and S2).

**Author Contributions**

V.V.P. and L.T.M. designed the study. V.V.P. guided all aspects of system development, sample collection and processing, analyzed the results and wrote the manuscript. A.M.S. made the $^{14}$C measurements. E.M.C. built the air sampler. A.C. collected the air samples. E.M.C, R.K. and P.P. processed the air samples. B.Y. and W.H. graphitized the samples. 395 All authors contributed to improving the manuscript.

### Competing Interests

The authors declare that they have no conflict of interest.

**Acknowledgements**

This work was supported by the David and Lucille Packard Fellowship for Science and Engineering (to Petrenko). We acknowledge the financial support from the Australian Government for the Centre for Accelerator Science at ANSTO through the National 405 Collaborative Research Infrastructure Strategy. We thank Ed Dlugokencky, Brian Vasel and Darryl Kuniyuki for facilitating access to sampling at MLO, and Emily Mesiti for assistance with researching and ordering system components.

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

micro-sample C-14 AMS at the ANTARES AMS facility, Nuclear Instruments &
Methods in Physics Research Section B-Beam Interactions with Materials and
Atoms, 268, 919-923, 10.1016/j.nimb.2009.10.064, 2010.

Spivakovsky, C. M., Logan, J. A., Montzka, S. A., Balkanski, Y. J., Foreman-Fowler, M.,
Jones, D. B. A., Horowitz, L. W., Fusco, A. C., Brenninkmeijer, C. A. M.,
Prather, M. J., Wofsy, S. C., and McElroy, M. B.: Three-dimensional
climatological distribution of tropospheric OH: Update and evaluation, Journal of
Geophysical Research, 105, 8931-8980, 2000.

Stuiver, M., and Polach, H. A.: Reporting of C-14 Data - Discussion, Radiocarbon, 19,
355-363, 1977.



Vimont, I. J., Turnbull, J. C., Petrenko, V. V., Place, P. F., Karion, A., Miles, N. L., Richardson, S. J., Gurney, K., Patarasuk, R., Sweeney, C., Vaughn, B., and

White, J. W. C.: Carbon monoxide isotopic measurements in Indianapolis constrain urban source isotopic signatures and support mobile fossil fuel emissions as the dominant wintertime CO source, Elementa-Sci Anthrop, 5, ARTN 63, 10.1525/elementa.136, 2017.

Volz, A., Ehhalt, D. H., and Derwent, R. G.: Seasonal and Latitudinal Variation of 14CO

and the Tropospheric Concentration of OH Radicals, Journal of Geophysical Research, 86, 5163-5171, 1981.

Voulgarakis, A., Naik, V., Lamarque, J. F., Shindell, D. T., Young, P. J., Prather, M. J., Wild, O., Field, R. D., Bergmann, D., Cameron-Smith, P., Cionni, I., Collins, W. J., Dalsoren, S. B., Doherty, R. M., Eyring, V., Faluvegi, G., Folberth, G. A.,

Horowitz, L. W., Josse, B., MacKenzie, I. A., Nagashima, T., Plummer, D. A., Righi, M., Rumbold, S. T., Stevenson, D. S., Strode, S. A., Sudo, K., Szopa, S., and Zeng, G.: Analysis of present day and future OH and methane lifetime in the ACCMIP simulations, Atmospheric Chemistry and Physics, 13, 2563-2587, Doi 10.5194/Acp-13-2563-2013, 2013.

Weinstock, B.: Carbon Monoxide - Residence Time in Atmosphere, Science, 166, 224-225, DOI 10.1126/science.166.3902.224, 1969.

Yang, B., and Smith, A. M.: Conventionally Heated Microfurnace for the Graphitization of Microgram-Sized Carbon Samples, Radiocarbon, 59, 859-873, 10.1017/Rdc.2016.89, 2017.






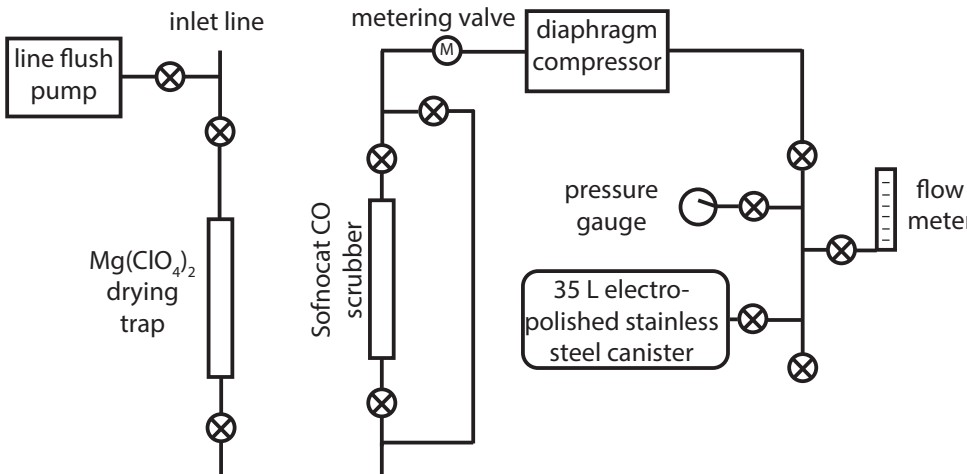


Figure 1. Schematic of the new atmospheric $^{14}$CO sampling system deployed at the Mauna Loa Observatory. An "X" within a circle denotes a valve (Swagelok, 4H bellows-sealed).




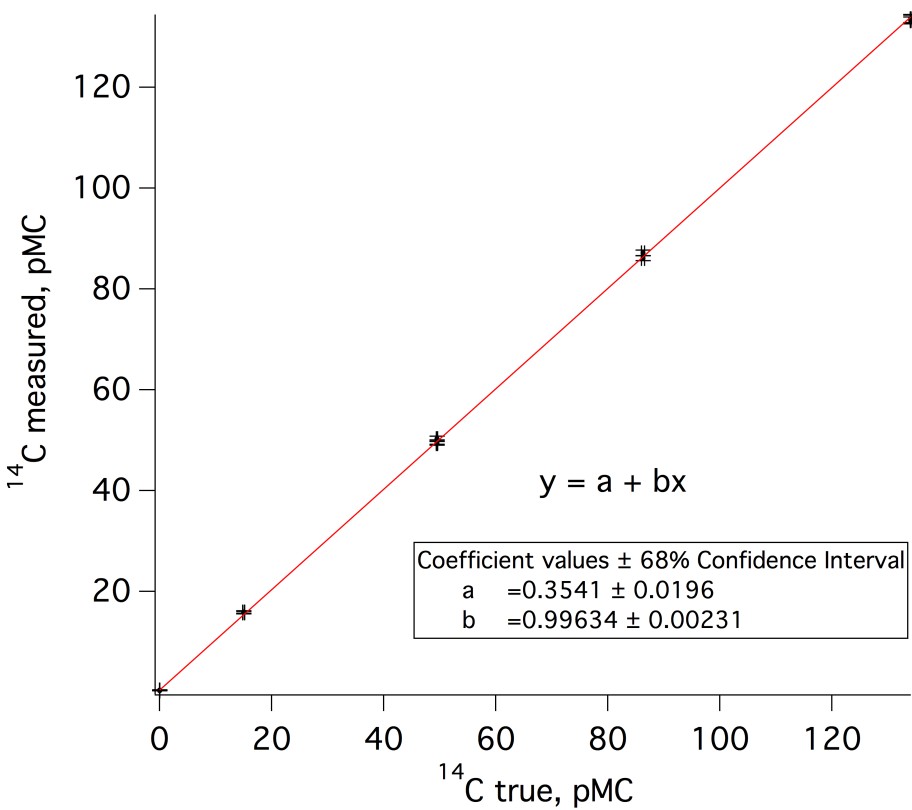

Figure 2. A plot of measured versus true (accepted) $^{14}$C values for commensurately-sized $^{14}$C standards and blanks that were processed at ANSTO concurrently with the second set of MLO $^{14}$CO samples and blanks (Samples 7 – 18 in Table S1 and Blanks 3 – 6 in Table S2). The data point clusters, going from left to right, represent a previously-characterized cylinder of $^{14}$C-depleted $CO_2$ ($^{14}$C true = 0.03 pMC), IAEA-C8 ($^{14}$C true = 15.03 pMC), IAEA-C7 ($^{14}$C true = 49.53 pMC), a second previously-characterized cylinder of $CO_2$ ($^{14}$C true = 86.27 pMC) and HOxII ($^{14}$C true = 134.06 pMC).





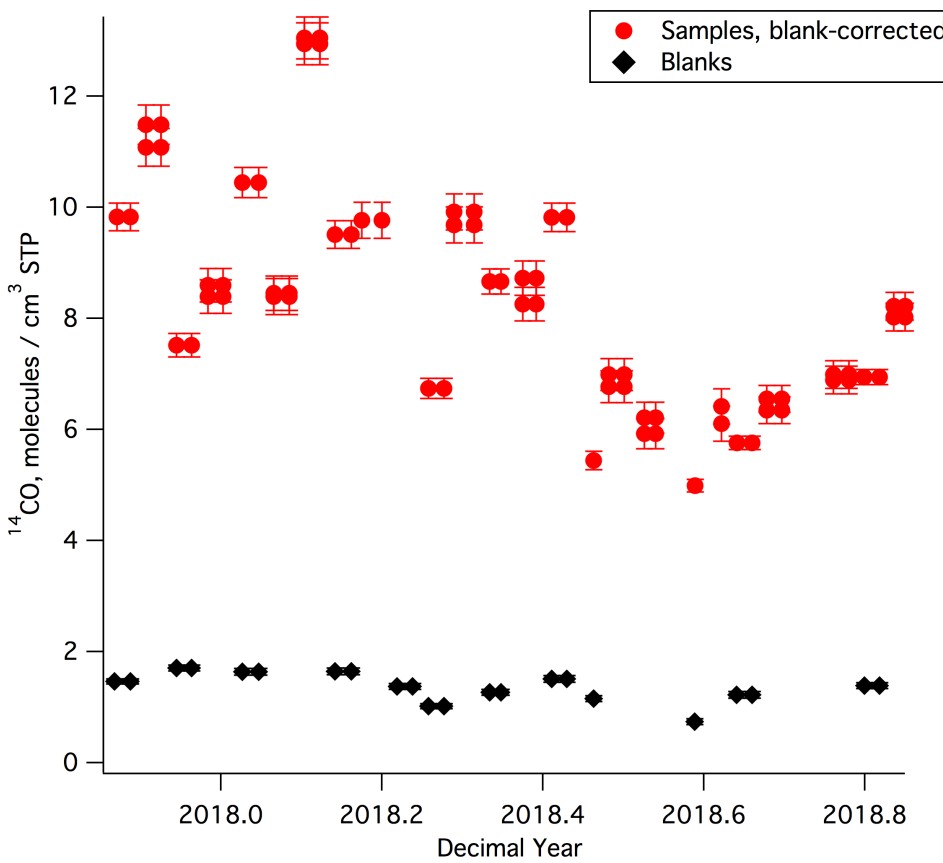

Figure 3. [$^{14}$CO] results for all MLO samples and blanks. Most samples and blanks were collected by half-filling the canisters on 2 separate days. To illustrate this, [$^{14}$CO] values for these samples and blanks are plotted for each of these dates, appearing twice as adjacent data points. All shown [$^{14}$CO] uncertainties are 1 σ.






| Sample subset 1 | N | Mean 1 σ uncertainty, as % of value | Sample subset 2 | N | Mean 1 σ uncertainty, as % of value | Can null hypothesis be rejected at 5% significance level? | p |
|---|---|---|---|---|---|---|---|
| All ≈22 µgC | 25 | 3.3 | All ≈50 µgC | 11 | 3.4 | NO | 0.72 |
| All accompanied by blanks | 11 | 2.5 | All not accompanied by blanks | 25 | 3.7 | YES | $1.2 \times 10^{-6}$ |
| ≈22 µgC not accompanied by blanks | 17 | 3.6 | ≈50 µgC not accompanied by blanks | 8 | 3.9 | NO | 0.29 |
| ≈22 µgC accompanied by blanks | 8 | 2.7 | ≈50 µgC accompanied by blanks | 3 | 2.1 | YES | $8.4 \times 10^{-4}$ |
| ≈22 µgC not accompanied by blanks | 17 | 3.6 | ≈22 µgC accompanied by blanks | 8 | 2.7 | YES | $7.4 \times 10^{-5}$ |
| ≈50 µgC not accompanied by blanks | 8 | 3.9 | ≈50 µgC accompanied by blanks | 3 | 2.1 | YES | $4.9 \times 10^{-3}$ |

Table 1. Results of a two-sample t-test investigating the effects of measured sample mass, whether the sample was accompanied by a blank, or both on the final relative uncertainty in the determined sample [$^{14}$CO] value. N is the number of samples in a particular subset. The null hypothesis is that the two subsets being compared are drawn from populations with equal means. The null hypothesis is rejected (i.e., the t-test indicates that the means of the subsets are significantly different) if the probability (p) of the observed subsets occurring when the underlying populations have equal means is less than 0.05 (< 5%).

