# Peer review of "An improved method for atmospheric $^{14}\text{CO}$ measurements"

_Atmospheric Measurement Techniques, 2020_

## Referee Comment (RC1) · Anonymous Referee #1 · 9 Oct 2020

General comments:

This manuscript describes an improved method for the collection of atmospheric samples used for the determination of $^{14}CO$ concentration, which serves as a useful tracer in characterizing the variability of atmospheric hydroxyl radical concentration. Since CO is present only in trace quantities in atmospheric samples, isotopic measurements, especially $^{14}CO$ measurements demand collection of larger air samples in order to enable measurements with acceptable uncertainties. Such large volume samplings can be both logistically challenging and expensive. Further, performing radiocarbon measurements on small samples (10-50 µgC) poses additional challenge both during graphitization and measurement. Through the methods described in this manuscript, following solutions have been presented: 1) use of a logistically attractive sample volume, 2) amplifying the mass of carbon present in the sample through dilution with high CO containing air to enable more precise measurements than possible in earlier work and 3) demonstrates the importance and the need of procedural blank sampling together with the actual sample collection.

The manuscript is very well written and falls within the scope of the journal AMT. I would recommend this manuscript for publication with some very minor clarifications.

Specific comments:

1. Page 6 Line 167: What pressures do you use during the "pressure-flush" step?

2. Page 6 Line 181: The use of italicized Latin forms should be consistent throughout the manuscript (see page 5 line 138).

3. Page 7 Line 196: Please specify the amount of gas used up during the CRDS measurement.

4. Page 7 Line 197: Was this $^{14}C$-depleted high CO-in-air prepared in-house or purchased through a commercial vendor?

5. Page 7 Line 218: Please provide a part number/manufacturers details if purchased through a commercial vendor.

6. Page 11 Line 326-331: What part of this variability that you observe in your procedural blank could be due to memory from the canister itself? Do you clean the canisters in a special way and perform some sort of possible outgassing test? Could you please comment on this?

7. Figure 2: In a plot that covers a large dynamic range, it is common to display a residual to the fit which makes visualization of the distribution of your dataset around the fit very easy. Could you please include this?

8. Figure 3: If one looks at your data carefully, there is a noticeable correlation (although weak) between the $^{14}CO$ content measured in the blanks vs. the blank-corrected samples collected on the same day. Could you please comment on why this is the case?

---

## Referee Comment (RC2) · Martin Manning (Referee) · 23 Oct 2020

**Comments on Petrenko *et al* 2020, "An improved method for atmospheric $^{14}$CO measurements**

Martin Manning, New Zealand Climate Change Research Institute, Victoria University of Wellington

**General comments**

This paper gives a well organised summary of what is clearly a significant improvement in our ability to determine atmospheric oxidation rates by using the tracer $^{14}$CO. Some key points are:

- the quality of $^{14}$CO concentrations is now well established for air samples significantly smaller than have been used previously, e.g. the air samples used here are five to ten times smaller than used in other studies ;
- while some aspects of the sample treatment are similar to that done in previous studies, the description of the complete process from air collection to correction of AMS measurements is very well set out;
- recognition that "blank" samples stored in cylinders can have cosmogenic $^{14}$CO production continuing to occur inside them is a point that is only considered implicitly in other papers on this tracer;
- there is a thorough treatment of corrections and uncertainties in the final results and the quality of analysis is shown through admission that there are still some issues to be resolved, e.g. variation in blanks covered in lines 334 – 337.

My only significant concern with the paper is its very brief coverage of what is known about $^{14}$CO production rates. While the Kovaltsov et al, 2012, paper is cited, most readers will miss the point that this was a major advance by Ilya Usoskin's group as it has resolved a long-standing difference between model derived $^{14}$C production rates and estimates based on radiocarbon dating. Also, it was followed up by Poluianov *et al*, 2016 (see references below) which showed that a significant amount of $^{14}$C production occurs above the 10 hPa level in the atmosphere as has been expected by some experts in high energy physics, and has not been reflected at all in papers such as Masarik & Beer, 1999.

Similarly, Usoskin's group regularly update their estimates of monthly changes in the average cosmic ray modulation strength (Phi) which is the primary cause for changes in $^{14}$C production rate. See http://cosmicrays.oulu.fi/phi/phi.html and http://cosmicrays.oulu.fi/phi/Phi_Table_2017.txt. This data source could be used to quantify the level of agreement between periods 1996-97 and 2017-18 that are used in section 3.

Despite these comments I would recommend that this paper be published after the authors have considered some suggestions made below.

**Specific comments**

line 88: As noted above, I would recommend that this sentence be expanded to cover the two references Kovaltsov *et al* and Poluianov *et al* which have set out much more detailed estimates for $^{14}$C production rates and their spatial distribution.

lines 96-97: determination of a global average $^{14}$C production rate needs global coverage for data on the solar modulation of cosmic ray activity. I would recommend Usoskin *et al*, 2011, (see below) as a reference to be added here.

line 98: this is a minor point but there are other estimates of the $^{14}$CO production yield, e.g. by Jöckel and Brenninkmeijer, and these vary over a small range of about 93 – 96%. It is another small

source of uncertainty as it can vary with altitude and mean the vertical distribution of $^{14}CO$ production is not quite the same as $^{14}C$ production.

lines 150 – 291: while there may be more detail in this section than some readers will follow, I would like to say that it is a very good summary of the range of issues that have to be dealt with in order to have precision in the results.

lines 184 – 187: presumably records are kept of the flight used to transport the sample from Honolulu, but do these use the same type of aircraft and so are expected to be at similar altitudes during the flight. Also have there been any estimates of in situ $^{14}CO$ production during shipping to the University of Rochester by doing repeated shipping of a blank test sample? And will the storage time at Rochester vary between samples?

lines 266 – 267: as mentioned in my general comments, I think this is a very important point.

lines 304 – 311: to quantify my general comments on comparing periods 1996-97 and 2017-18, http://cosmicrays.oulu.fi/phi/phi.html shows that the cosmogenic modulation potential averaged over 1996 – 1997 was 506 MV and over 2017-18 was 456 MV. The weaker modulation effect in 2017-18 increases the global average production rate by 4% when the Kovaltsov *et al* production rates are used, and the Poluianov *et al* rates have very similar global averages.

lines 304 – 311 again: while MLO and Ragged Point Barbados have similar latitudes their altitudes are different and local cosmogenic $^{14}CO$ production rates will be about 20 times larger at MLO. This is well recognised by rapid removal of the MLO samples to lower altitudes but also leaves a question about comparing the atmospheric observations at different altitudes. So, I would suggest adding the point that this comparison is valid because rapid vertical mixing in the troposphere means there are only small vertical gradients in $^{14}CO$ concentrations.

331 – 337: does this comparison of the two different values for blanks lead to a conclusion?

374 – 377: following on from that last question, have surface effects in the canisters been considered and have they been treated to avoid variations in forms of carbon becoming attached to the interior surface?

**References**

Poluianov, S.V., Kovaltsov, G.A., Mishev, A.L., and Usoskin, I.G., 2016: Production of cosmogenic isotopes $^7Be$, $^{10}Be$, $^{14}C$, $^{22}Na$, and $^{36}Cl$ in the atmosphere: Altitudinal profiles of yield functions. Journal of Geophysical Research: Atmospheres, 121, 8125-8136.

Usoskin, I.G., Bazilevskaya, G.A., and Kovaltsov, G.A., 2011: Solar modulation parameter for cosmic rays since 1936 reconstructed from ground-based neutron monitors and ionization chambers. J. Geophys. Res, 116, A02104, doi:10.1029/2010JA016105.

---

## Author Comment (AC1) · 4 Dec 2020

We would like to thank this reviewer for their careful reading of the manuscript and their helpful and constructive comments. The reviewer comments are shown below in *blue italics*, with our responses in regular black font.

*Comments on "An improved method for atmospheric 14CO measurements" by Petrenko et al.*

*General comments:*

*This manuscript describes an improved method for the collection of atmospheric samples used for the determination of $^{14}$CO concentration, which serves as a useful tracer in characterizing the variability of atmospheric hydroxyl radical concentration. Since CO is present only in trace quantities in atmospheric samples, isotopic measurements, especially $^{14}$CO measurements demand collection of larger air samples in order to enable measurements with acceptable uncertainties. Such large volume samplings can be both logistically challenging and expensive. Further, performing radiocarbon measurements on small samples (10-50 µgC) poses additional challenge both during graphitization and measurement. Through the methods described in this manuscript, following solutions have been presented: 1) use of a logistically attractive sample volume, 2) amplifying the mass of carbon present in the sample through dilution with high CO containing air to enable more precise measurements than possible in earlier work and 3) demonstrates the importance and the need of procedural blank sampling together with the actual sample collection.*

*The manuscript is very well written and falls within the scope of the journal AMT. I would recommend this manuscript for publication with some very minor clarifications.*

*Specific comments:*
*1. Page 6 Line 167: What pressures do you use during the "pressure-flush" step?*

≈25 psig (this is somewhat variable as the pressure builds very quickly when the vent valve is closed). This detail will be added to the revised manuscript.

*2. Page 6 Line 181: The use of italicized Latin forms should be consistent throughout the manuscript (see page 5 line 138).*

All instances of "in situ" will be italicized in the revised manuscript

*3. Page 7 Line 196: Please specify the amount of gas used up during the CRDS measurement.*

This is ≈800 cm$^3$ STP; we will add this information to the revised manuscript

*4. Page 7 Line 197: Was this $^{14}$C-depleted high CO-in-air prepared in-house or purchased through a commercial vendor?*

This custom gas mixture was purchased from Praxair. We will include this detail in the revised manuscript.

*5. Page 7 Line 218: Please provide a part number/manufacturers details if purchased through a commercial vendor.*

This was Schimadzu part no. 630-00996-00, we will add this detail to the revised manuscript

*6. Page 11 Line 326-331: What part of this variability that you observe in your procedural blank could be due to memory from the canister itself? Do you clean the canisters in a special way and perform some sort of possible outgassing test? Could you please comment on this?*

The stainless steel canisters have been electropolished at the time of manufacturing, which helps to clean and passivate the surface; the fact that the canisters are electropolished is already mentioned in the manuscript (end of 1st paragraph in section 2.1). Prior to being reused, the canisters are evacuated to 0.25 torr and leak-tested overnight. The best indicator that we have for a lack of significant "memory" from the canisters themselves is the consistently low CO mole fraction measured in the blanks ($3.7 \pm 1.8$ ($1\sigma$) nmol mol$^{-1}$ see also response to point 8 below). Following the dilutions with the high-CO, $^{14}$C-depleted gas, the mean CO mole fraction in the sample and blank canisters was $512 \pm 36$ ($1\sigma$) nmol mol$^{-1}$ for the ≈22 µgC samples and $1134 \pm 19$ ($1\sigma$) nmol mol$^{-1}$ for the ≈50 µgC samples. Assuming that the observed CO in the blanks is originating from canister "memory", this memory would represent <1 % of the CO present in the canister prior to the evacuation. Further, following the dilution the $^{14}$C activity of CO in the sample canisters is much lower than that of typical atmospheric CO. Assuming 3.7 nmol mol$^{-1}$ of CO with a typical (after dilutions for ≈22 µgC samples) $^{14}$C activity of 60 pMC is added via canister "memory", this translates to 0.07 $^{14}$CO molecules / cm$^3$ STP – which is much smaller than the variability between the blanks and similar to the estimated $1\sigma$ uncertainty for blank $^{14}$CO (see Table S2 in the original manuscript).

That said, it is much more likely that the small amount of CO observed in the blanks is due to a combination of CO outgassing from the KNF N145 pump used in the sampling system and from the sample canisters. The observed blank CO mole fractions are consistent with those expected based on sampling system and canister tests conducted in our laboratory prior to this and other projects that used the same equipment. Blank 13 was the only blank for which the preceding sample in the same canister was a ≈50 µgC sample with calculated CO mole fraction of 1112 nmol mol$^{-1}$ following the dilution; for other blanks the preceding sample or blank in the same canister was ≈22 µgC in size with diluted CO mole fractions of ≈500 nmol mol$^{-1}$. CO mole fraction measured in Blank 13 (3.6 nmol mol$^{-1}$) is not anomalous compared to other blanks, arguing against a canister CO memory effect.

The sampling canisters outgas CO at a rate of 1 – 3 nmol mol$^{-1}$ per month as determined in tests associated with prior projects. However, again, CO outgassing at this rate would not affect the sample $^{14}$CO results significantly.

In the revised manuscript, we will add information regarding canister evacuation in between samples, as well as a brief statement that the consistently low CO mole fraction in the blanks rules out the possibility of significant $^{14}$CO interference from canister memory or outgassing. We will also add measured CO mole fractions for the blanks to Table S2.

*7. Figure 2: In a plot that covers a large dynamic range, it is common to display a residual to the fit which makes visualization of the distribution of your dataset around the fit very easy. Could you please include this?*

This will be added to Figure 2 in the revised manuscript.

*8. Figure 3: If one looks at your data carefully, there is a noticeable correlation (although weak) between the $^{14}CO$ content measured in the blanks vs. the blank-corrected samples collected on the same day. Could you please comment on why this is the case?*

[Figure]

*Figure R1. Observed $^{14}CO$ correlation for blank-sample pairs collected and analyzed as part of the Mauna Loa $^{14}CO$ campaign. This correlation appears to be significant, with a p value of 0.007.*

We agree that this correlation is puzzling (see figure R1 above), but it cannot be due to analytical artifacts, for the following reasons. One analytical problem that could in principle result in such a correlation would be a failure of the Sofnocat 423 reagent (see Figure 1 in manuscript) to fully oxidize all CO (and $^{14}CO$) in the sampled air when sampling is performed in blank mode. In this case, the blank-sample $^{14}CO$ relationship in Figure R1 suggests that ≈11% of sample CO (and $^{14}CO$) breaks through the Sofnocat CO scrubber. However, this is ruled out by the consistently low CO mole fraction in the blanks (see response to point 6 above) that is not positively correlated to the CO mole fraction in the samples collected on the same days (see Figure R2 below).

[Figure]

*Figure R2. Comparison of measured CO mole fraction for blank-sample pairs collected and analyzed as part of the Mauna Loa $^{14}$CO campaign. While the data suggest a negative slope for the dependence of blank CO mole fraction on sample CO mole fraction, in this case the correlation does not appear to be significant, with a p value of 0.17.*

The possibility of $^{14}$CO in the blanks being significantly affected by "memory" in the sampling canisters was already discussed and ruled out in the response to reviewer's point 6 above. We also considered the possibility that the correlation could be due to carbon memory in the air processing system at the U Rochester laboratory. A very similar system at the National Institute for Water and Atmospheric Research (NIWA) in Wellington, New Zealand utilizing similar components (including the same type of platinized quartz wool) has been previously demonstrated to be free of memory artifacts when operated in $CH_4$ mode (Petrenko et al., 2008). To examine whether any carbon memory might exist in the U Rochester system operated in CO mode, we compared measured $^{14}$CO for sample-sample pairs collected on the same days (values for all samples were already shown in Table S1). There are six such pairs where one of the samples was processed on the system following a sample, and another following a blank. If the system does indeed have a memory, we would expect lower $^{14}$CO for samples that were processed following a blank. The average $^{14}$CO offset between such pairs is 0.03 molecules / cm$^3$ STP, while the standard deviation of the offsets is 0.35 molecules / cm$^3$ STP. We thus conclude that there is no evidence for a significant memory effect in the U Rochester air processing system.

We can also rule out memory effects in the micro-conventional furnaces used to graphitize the sample-derived $CO_2$ at ANSTO based on tests conducted on these furnaces (Yang and Smith, 2017).

Based on all of the above, we can rule out the possibility that the $^{14}$CO correlation observed for blank-sample pairs is due to analytical artifacts. We further note that $^{14}$CO concentrations observed in blanks 9 and 10 (1.15 and 0.74 molecules / cm$^3$ STP) are similar to prior estimates of in situ $^{14}$CO production from a jet aircraft flight (0.9 molecules / cm$^3$ STP, with a ≈30% uncertainty; Lowe et al., 2002). Blanks 9 and 10 were filled in a single day, transported to sea level within hours and shipped to U Rochester the following day; thus $^{14}$CO in these blanks likely represents only the in situ $^{14}$CO from aircraft transport.

Unfortunately, we do not at this point have a clear explanation for the correlation. It may be possible that this effect is related to airplane trajectories being influenced by atmospheric conditions. Lower atmospheric $^{14}CO$ at Mauna Loa is generally associated with warmer low-latitude air masses. It may be possible that in such conditions, the airplanes that transport our samples and blanks from Hawaii to Rochester fly at cruising altitudes corresponding to somewhat higher pressures (to maintain constant air density in warmer air). This would result in lower in situ $^{14}CO$ production rates in the tanks during airplane transport. Unfortunately, FedEx (the carrier for all our samples) does not provide routing information for past shipments, so we are unable to verify this hypothesis.

In the revised manuscript, we will mention the $^{14}CO$ correlation in the blank-sample pairs and include the detailed above discussion in the supplement.

References:

Yang, B. and Smith, A.M., 2017. Conventionally heated microfurnace for the graphitization of microgram-sized carbon samples. *Radiocarbon*, 59, 859 – 873.

---

## Author Response (AR1)

**Response to Reviews of Petrenko et al., "An improved method for atmospheric $^{14}$CO measurements"**

We would first like to thank the reviewers for their careful reading of the manuscript and their helpful and constructive comments. The reviewer comments are shown below in *blue italics*, with our responses in regular black font.

**Reviewer 1**

*Comments on "An improved method for atmospheric 14CO measurements" by Petrenko et al.*

*General comments:*

*This manuscript describes an improved method for the collection of atmospheric samples used for the determination of $^{14}$CO concentration, which serves as a useful tracer in characterizing the variability of atmospheric hydroxyl radical concentration. Since CO is present only in trace quantities in atmospheric samples, isotopic measurements, especially $^{14}$CO measurements demand collection of larger air samples in order to enable measurements with acceptable uncertainties. Such large volume samplings can be both logistically challenging and expensive. Further, performing radiocarbon measurements on small samples (10-50 µgC) poses additional challenge both during graphitization and measurement. Through the methods described in this manuscript, following solutions have been presented: 1) use of a logistically attractive sample volume, 2) amplifying the mass of carbon present in the sample through dilution with high CO containing air to enable more precise measurements than possible in earlier work and 3) demonstrates the importance and the need of procedural blank sampling together with the actual sample collection.*

*The manuscript is very well written and falls within the scope of the journal AMT. I would recommend this manuscript for publication with some very minor clarifications.*

*Specific comments:*
*1. Page 6 Line 167: What pressures do you use during the "pressure-flush" step?*

≈25 psig (this is somewhat variable as the pressure builds very quickly when the vent valve is closed). This detail has been added to the revised manuscript (p. 6, line 171).

*2. Page 6 Line 181: The use of italicized Latin forms should be consistent throughout the manuscript (see page 5 line 138).*

All instances of "in situ" have been italicized in the revised manuscript.

*3. Page 7 Line 196: Please specify the amount of gas used up during the CRDS measurement.*

This is ≈800 cm$^3$ STP; this information has been added to the revised manuscript (p. 6, line 200).

*4. Page 7 Line 197: Was this $^{14}$C-depleted high CO-in-air prepared in-house or purchased through a commercial vendor?*

This custom gas mixture was purchased from Praxair. This detail has been included in the revised manuscript (p. 6, line 201).

*5. Page 7 Line 218: Please provide a part number/manufacturers details if purchased through a commercial vendor.*

This was Schimadzu part no. 630-00996-00, this detail has been added to the revised manuscript (p.7, line 222).

*6. Page 11 Line 326-331: What part of this variability that you observe in your procedural blank could be due to memory from the canister itself? Do you clean the canisters in a special way and perform some sort of possible outgassing test? Could you please comment on this?*

The stainless steel canisters have been electropolished at the time of manufacturing, which helps to clean and passivate the surface; the fact that the canisters are electropolished is already mentioned in the manuscript (end of 1$^{st}$ paragraph in section 2.1). Prior to being reused, the canisters are evacuated to 0.25 torr and leak-tested overnight. The best indicator that we have for a lack of significant "memory" from the canisters themselves is the consistently low CO mole fraction measured in the blanks (3.7 ± 1.8 (1$\sigma$) nmol mol$^{-1}$ see also response to point 8 below). Following the dilutions with the high-CO, $^{14}$C-depleted gas, the mean CO mole fraction in the sample and blank canisters was 512 ± 36 (1$\sigma$) nmol mol$^{-1}$ for the ≈22 µgC samples and 1134 ± 19 (1$\sigma$) nmol mol$^{-1}$ for the ≈50 µgC samples. Assuming that the observed CO in the blanks is originating from canister "memory", this memory would represent <1 % of the CO present in the canister prior to the evacuation. Further, following the dilution the $^{14}$C activity of CO in the sample canisters is much lower than that of typical atmospheric CO. Assuming 3.7 nmol mol$^{-1}$ of CO with a typical (after dilutions for ≈22 µgC samples) $^{14}$C activity of 60 pMC is added via canister "memory", this translates to 0.07 $^{14}$CO molecules / cm$^3$ STP – which is much smaller than the variability between the blanks and similar to the estimated 1$\sigma$ uncertainty for blank $^{14}$CO (see Table S2 in the original manuscript).

That said, it is much more likely that the small amount of CO observed in the blanks is due to a combination of CO outgassing from the KNF N145 pump used in the sampling system and from the sample canisters. The observed blank CO mole fractions are consistent with those expected based on sampling system and canister tests conducted in our laboratory prior to this and other projects that used the same equipment. Blank 13 was the only blank for which the preceding sample in the same canister was a ≈50 µgC sample with calculated CO mole fraction of 1112 nmol mol$^{-1}$ following the dilution; for other blanks the preceding sample or blank in the same canister was ≈22 µgC in size with diluted CO mole fractions of ≈500 nmol mol$^{-1}$. CO mole fraction measured in Blank 13 (3.6 nmol mol$^{-1}$) is not anomalous compared to other blanks, arguing against a canister CO memory effect.

The sampling canisters outgas CO at a rate of 1 – 3 nmol mol$^{-1}$ per month as determined in tests associated with prior projects. However, again, CO outgassing at this rate would not affect the sample $^{14}$CO results significantly.

In the revised manuscript, we added the pressure to which canisters were evacuated in between samples (p. 5, line 164). We also added a brief statement (p.11, line 338) that clarifies that the $^{14}$CO blank is not arising from outgassing or analytical artifacts and points to a more detailed discussion in the Supplement. Finally, we have added a section in the Supplement that discusses possible effects of outgassing and "memory" from sample canisters on blank $^{14}$CO. In support of this discussion, CO mole fractions measured in the blanks were also added to Table S2.

*7. Figure 2: In a plot that covers a large dynamic range, it is common to display a residual to the fit which makes visualization of the distribution of your dataset around the fit very easy. Could you please include this?*

This has been added to Figure 2 in the revised manuscript.

*8. Figure 3: If one looks at your data carefully, there is a noticeable correlation (although weak) between the $^{14}CO$ content measured in the blanks vs. the blank-corrected samples collected on the same day. Could you please comment on why this is the case?*

We agree that this correlation is puzzling (see Figure S1 in the revised Supplement), but it cannot be due to analytical artifacts, for the following reasons. One analytical problem that could in principle result in such a correlation would be a failure of the Sofnocat 423 reagent (see Figure 1 in manuscript) to fully oxidize all CO (and $^{14}CO$) in the sampled air when sampling is performed in blank mode. In this case, the blank-sample $^{14}CO$ relationship in Figure S1 suggests that ≈12% of sample CO (and $^{14}CO$) breaks through the Sofnocat CO scrubber. However, this is ruled out by the consistently low CO mole fraction in the blanks (see response to point 6 above) that is not positively correlated to the CO mole fraction in the samples collected on the same days (see Figure S2 in the revised Supplement).

The possibility of $^{14}CO$ in the blanks being significantly affected by "memory" in the sampling canisters was already discussed and ruled out in the response to reviewer's point 6 above. We also considered the possibility that the correlation could be due to carbon memory in the air processing system at the U Rochester laboratory. A very similar system at the National Institute for Water and Atmospheric Research (NIWA) in Wellington, New Zealand utilizing similar components (including the same type of platinized quartz wool) has been previously demonstrated to be free of memory artifacts when operated in $CH_4$ mode (Petrenko et al., 2008). To examine whether any carbon memory might exist in the U Rochester system operated in CO mode, we compared measured $^{14}CO$ for sample-sample pairs collected on the same days (values for all samples were already shown in Table S1). There are six such pairs where one of the samples was processed on the system following a sample, and another following a blank. If the system does indeed have a memory, we would expect lower $^{14}CO$ for samples that were processed following a blank. The average $^{14}CO$ offset between such pairs is 0.03 molecules / $cm^3$ STP, while the standard deviation of the offsets is 0.35 molecules / $cm^3$ STP. We thus conclude that there is no evidence for a significant memory effect in the U Rochester air processing system.

We can also rule out memory effects in the micro-conventional furnaces used to graphitize the sample-derived $CO_2$ at ANSTO based on tests conducted on these furnaces (Yang and Smith, 2017).

Based on all of the above, we can rule out the possibility that the $^{14}CO$ correlation observed for blank-sample pairs is due to analytical artifacts. We further note that $^{14}CO$ concentrations observed in blanks 9 and 10 (1.15 and 0.74 molecules / $cm^3$ STP) are similar to prior estimates of in situ $^{14}CO$ production from a jet aircraft flight (0.9 molecules / $cm^3$ STP, with a ≈30% uncertainty; Lowe et al., 2002). Blanks 9 and 10 were filled in a single day, transported to sea level within hours and shipped to U Rochester the following day; thus [14]CO in these blanks likely represents only the in situ [14]CO from aircraft transport.

Unfortunately, we do not at this point have a clear explanation for the correlation. It may be possible that this effect is related to airplane trajectories being influenced by atmospheric conditions. Lower atmospheric [14]CO at Mauna Loa is generally associated with warmer low-latitude air masses. It may be possible that in such conditions, the airplanes that transport our samples and blanks from Hawaii to Rochester fly at cruising altitudes corresponding to somewhat higher pressures (to maintain constant air density in warmer air). This would result in lower in situ [14]CO production rates in the tanks during airplane transport. Unfortunately, FedEx (the carrier for all our samples) does not provide routing information for past shipments, so we are unable to verify this hypothesis.

In the revised manuscript, we have mentioned the [14]CO correlation in the blank-sample pairs in the Figure 3 caption and referred to the detailed discussion in the revised Supplement. In the revised Supplement, we have added a detailed discussion of this correlation and Figures S1 and S2. We have also added CO mole fractions for samples and blanks into Tables S1 and S2, respectively.

**Dr. Martin Manning**

*Comments on Petrenko et al 2020, "An improved method for atmospheric [14]CO measurements*

*Martin Manning, New Zealand Climate Change Research Institute, Victoria University of Wellington*

*General comments*

*This paper gives a well organised summary of what is clearly a significant improvement in our ability to determine atmospheric oxidation rates by using the tracer [14]CO. Some key points are:*

- *the quality of [14]CO concentrations is now well established for air samples significantly smaller than have been used previously, e.g. the air samples used here are five to ten times smaller than used in other studies ;*
- *while some aspects of the sample treatment are similar to that done in previous studies, the description of the complete process from air collection to correction of AMS measurements is very well set out;*
- *recognition that "blank" samples stored in cylinders can have cosmogenic [14]CO production continuing to occur inside them is a point that is only considered implicitly in other papers on this tracer;*
- *there is a thorough treatment of corrections and uncertainties in the final results and the quality of analysis is shown through admission that there are still some issues to be resolved, e.g. variation in blanks covered in lines 334 – 337.*

*My only significant concern with the paper is its very brief coverage of what is known about [14]CO production rates. While the Kovaltsov et al, 2012, paper is cited, most readers will miss*

*the point that this was a major advance by Ilya Usoskin's group as it has resolved a long-standing difference between model derived $^{14}C$ production rates and estimates based on radiocarbon dating. Also, it was followed up by Poluianov et al, 2016 (see references below) which showed that a significant amount of $^{14}C$ production occurs above the 10 hPa level in the atmosphere as has been expected by some experts in high energy physics, and has not been reflected at all in papers such as Masarik & Beer, 1999.*

In the revised manuscript, we have added the Poluianov et al (2016) reference (p.3, line 88). We note that the main focus of this paper is on the analytical techniques, rather than on interpretation of the $^{14}CO$ results and their implications for atmospheric OH and the $^{14}C$ production scheme used in models. Considering this, we would prefer to keep the discussion of atmospheric $^{14}C$ production relatively brief.

*Similarly, Usoskin's group regularly update their estimates of monthly changes in the average cosmic ray modulation strength (Phi) which is the primary cause for changes in $^{14}C$ production rate. See http://cosmicrays.oulu.fi/phi/phi.html and http://cosmicrays.oulu.fi/phi/Phi_Table_2017.txt. This data source could be used to quantify the level of agreement between periods 1996-97 and 2017-18 that are used in section 3.*

We thank Dr. Manning for pointing this out. However, again, we would prefer to keep the focus of this manuscript on the analytical techniques. The qualitative comparison to prior Barbados measurements is used in the manuscript to support the overall argument that our technique produces reliable results. An in-depth quantitative comparison would require the consideration of changes in atmospheric $^{14}C$ production as Dr. Manning points out, as well as a chemistry – transport model. We feel that such an analysis is beyond the scope of this paper.

*Despite these comments I would recommend that this paper be published after the authors have considered some suggestions made below.*

*Specific comments*

*line 88: As noted above, I would recommend that this sentence be expanded to cover the two references Kovaltsov et al and Poluianov et al which have set out much more detailed estimates for $^{14}C$ production rates and their spatial distribution.*

These two references have been added in the revised manuscript (p.3, lines 87 – 88).

*lines 96-97: determination of a global average $^{14}C$ production rate needs global coverage for data on the solar modulation of cosmic ray activity. I would recommend Usoskin et al, 2011, (see below) as a reference to be added here.*

This reference has been be added to the revised manuscript (p.3, line 96).

*line 98: this is a minor point but there are other estimates of the $^{14}CO$ production yield, e.g. by Jöckel and Brenninkmeijer, and these vary over a small range of about 93 – 96%. It is another small source of uncertainty as it can vary with altitude and mean the vertical distribution of $^{14}CO$ production is not quite the same as $^{14}C$ production.*

While the Mak et al. (1994) study we cited used 93% for this value, some other studies have used a slightly different value of 95% (Jockel and Brenninkmeijer, 2002; Krol et al., 2008). In the revised manuscript, we have added the Jockel and Brenninkmeijer reference and give a 93 – 95% range for the $^{14}$CO yield (p.3, lines 97 – 98).

*lines 150 – 291: while there may be more detail in this section than some readers will follow, I would like to say that it is a very good summary of the range of issues that have to be dealt with in order to have precision in the results.*

Thanks!

*lines 184 – 187: presumably records are kept of the flight used to transport the sample from Honolulu, but do these use the same type of aircraft and so are expected to be at similar altitudes during the flight. Also have there been any estimates of in situ $^{14}$CO production during shipping to the University of Rochester by doing repeated shipping of a blank test sample? And will the storage time at Rochester vary between samples?*

Unfortunately, these records were not kept, and the routing / aircraft information for past shipments is not available from FedEx (the carrier for all our samples). See also the response to Reviewer 1 point 8. Our best estimates for in situ $^{14}$CO production in the canisters during Hawaii → Rochester shipping come from Blanks 9 and 10 (see Table S2), as these blanks were collected in a single day (rather than with a week in between canister half-fills, as was the case for most samples and blanks). These blanks yielded $^{14}$CO values of 1.15 and 0.74 molecules / cm$^3$ STP, and were already discussed in the original manuscript (middle paragraph on p. 11).

The storage time at Rochester is short (typically on the order of 1 week), but has varied by ± 1 week. However, the laboratory building is at an altitude of only ≈150 m a.s.l.. Further, the received sample and blank canisters are stored on the basement level of a 5-story building, which provides added shielding from cosmic rays. In situ $^{14}$CO production in sample canisters during storage at Rochester should therefore be negligible compared to in situ production during aircraft transport and the ≈1 week storage at the Mauna Loa observatory (3397 m a.s.l.).

*lines 266 – 267: as mentioned in my general comments, I think this is a very important point.*

*lines 304 – 311: to quantify my general comments on comparing periods 1996-97 and 2017-18, http://cosmicrays.oulu.fi/phi/phi.html shows that the cosmogenic modulation potential averaged over 1996 – 1997 was 506 MV and over 2017-18 was 456 MV. The weaker modulation effect in 2017-18 increases the global average production rate by 4% when the Kovaltsov et al production rates are used, and the Poluianov et al rates have very similar global averages.*

Please see our response regarding the Barbados – Mauna Loa results comparison in the general comments section above.

*lines 304 – 311 again: while MLO and Ragged Point Barbados have similar latitudes their altitudes are different and local cosmogenic $^{14}$CO production rates will be about 20 times larger at MLO. This is well recognised by rapid removal of the MLO samples to lower altitudes but also leaves a question about comparing the atmospheric observations at different altitudes. So, I would suggest adding the point that this comparison is valid because rapid vertical mixing in the troposphere means there are only small vertical gradients in $^{14}$CO concentrations.*

This again is a valid point. However, again, we are not attempting to do a detailed quantitative comparison of our Mauna Loa and prior Barbados $^{14}$CO results. Our preference would therefore be to leave such a detailed comparison (which would need to consider changes in atmospheric 14C production, site altitude, $^{14}$CO transport, etc) for a future study.

*331 – 337: does this comparison of the two different values for blanks lead to a conclusion?*

Yes, in the revised manuscript we have clarified that in situ $^{14}$CO production in the canisters during aircraft shipment from Hawaii to Rochester appears to be larger than in situ production during storage at MLO (p.11, line 346).

*374 – 377: following on from that last question, have surface effects in the canisters been considered and have they been treated to avoid variations in forms of carbon becoming attached to the interior surface?*

Please see the detailed response to point 6 from Reviewer 1, which posed very similar questions. Briefly, the consistently low CO mole fractions measured in the blanks indicate that any effects from the canisters (memory, outgassing) would be negligible for sample $^{14}$CO.

A few small typos were corrected.

[revised manuscript text omitted]